# The Role of the Inflammasome in Neurodegenerative Diseases

**DOI:** 10.3390/molecules26040953

**Published:** 2021-02-11

**Authors:** Federica Piancone, Francesca La Rosa, Ivana Marventano, Marina Saresella, Mario Clerici

**Affiliations:** 1IRCCS Fondazione Don Carlo Gnocchi, 20148 Milano, Italy; flarosa@dongnocchi.it (F.L.R.); imarventano@dongnocchi.it (I.M.); msaresella@dongnocchi.it (M.S.); mario.clerici@unimi.it (M.C.); 2Department of Pathophysiology and Transplantation, University of Milano, 20122 Milano, Italy

**Keywords:** inflammasome, NLRP3 inflammasome, neuroinflammation, neurodegenerative diseases, multiple sclerosis, Alzheimer’s disease, Parkinson’s diseases, amyotrophic lateral sclerosis

## Abstract

Neurodegenerative diseases are chronic, progressive disorders that occur in the central nervous system (CNS). They are characterized by the loss of neuronal structure and function and are associated with inflammation. Inflammation of the CNS is called neuroinflammation, which has been implicated in most neurodegenerative diseases, including Alzheimer’s disease (AD), Parkinson’s disease (PD), amyotrophic lateral sclerosis (ALS) and multiple sclerosis (MS). Much evidence indicates that these different conditions share a common inflammatory mechanism: the activation of the inflammasome complex in peripheral monocytes and in microglia, with the consequent production of high quantities of the pro-inflammatory cytokines IL-1β and IL-18. Inflammasomes are a group of multimeric signaling complexes that include a sensor Nod-like receptor (NLR) molecule, the adaptor protein ASC, and caspase-1. The NLRP3 inflammasome is currently the best-characterized inflammasome. Multiple signals, which are potentially provided in combination and include endogenous danger signals and pathogens, trigger the formation of an active inflammasome, which, in turn, will stimulate the cleavage and the release of bioactive cytokines including IL-1β and IL-18. In this review, we will summarize results implicating the inflammasome as a pivotal player in the pathogenesis of neurodegenerative diseases and discuss how compounds that hamper the activation of the NLRP3 inflammasome could offer novel therapeutic avenues for these diseases.

## 1. Introduction

Neuroinflammation plays a key role in the onset and the progression of several neurodegenerative diseases such as Alzheimer’s disease (AD), Parkinson’s disease (PD), multiple sclerosis (MS), and amyotrophic lateral sclerosis (ALS) [1,2,3].

Nevertheless, it has to be considered that, in primary neurodegenerative diseases characterized by the accumulation of misfolded proteins like AD and PD, it is not clear if inflammation might be the primary cause of disease or a reaction to pathology. Indeed, the pathophysiological hypothesis of neurodegenerative diseases relies on the fact that some proteins, changing their conformations, aggregate into fibrils or oligomers, resulting in neurotoxicity and leading to neurodegeneration and inflammation [4,5,6,7]. Neuroinflammation is a physiological response to exogenous and endogenous insults that target the central nervous system (CNS) and represents a protective response in the brain, but excessive inflammatory responses are detrimental to the CNS. Several inflammation-inducing stimuli, such as damage-associated molecular patterns (DAMPs) or pathogen-associated molecular patterns (PAMPs), are recognized by multiprotein complexes, called inflammasomes. This elicits a pro-inflammatory response mediated by the release of the inflammatory cytokines IL-1β and IL-18 [8]. The nucleotide-binding oligomerization domain leucine-rich repeat and pyrin domain-containing protein 3 (NLRP3) inflammasome, one of the most intensively investigated inflammasomes, has been reported to play a key role in neurodegenerative diseases.

Inflammasomes are a group of cytosolic multiprotein complexes that consist of a sensor molecule (NLR, AIM2-like receptors, ALR, and pyrin receptors), the adaptor apoptosis-associated speck-like protein, which contains a caspase recruitment domain (ASC), and pro-caspase-1 [9,10]. Inflammasome-inducing stimuli trigger the oligomerization of pattern recognition receptors (PRR) and the recruitment of pro-caspase-1 into the complex, leading to the generation of active caspase-1 that, consequently, will cleave inactive pro-peptides pro-IL-1β and pro-IL-18 into mature cytokines. Importantly, caspase-1 can also induce a pro-inflammatory form of cell death, pyroptosis, that features early plasma membrane rupture, thereby releasing the soluble intracellular fraction that fuels the inflammatory response [9,11]. NLRP3, as shown in Figure 1, is the best-characterized inflammasome; its activation involves a two-step process. A first signal, or “priming” signal, results in the NF-kB-dependent transcriptional upregulation of NLRP3 and pro-IL-1β, but also controls post-translational modifications of NLRP3 [12]. This initial trigger is followed by a second “activation” signal that induces the oligomerization and activation of the NLRP3 inflammasome. Besides this “canonical” NLRP3 inflammasome activation pathway, a “noncanonical” NLRP3 activatory pathway has been described. This pathway involves the activation of caspase-11 in mice (or its human orthologs caspase-4 and caspase-5) by cytosolic LPS, the induction of pyroptosis through the cleavage of gasdermin D (GSDMD), and the release of high mobility group box 1 protein (HMGB1), resulting in the production of IL-1β [9,13]. In both cases, the activation of NLRP3 inflammasome results in the cleavage of the pro-inflammatory IL-1β and IL-18; this leads to the generation of the biological active form of these proteins that initiates inflammatory signaling cascades, contributing to neuronal injury, cell death, and neuroinflammation [14,15].

We will briefly summarize data suggesting that the inflammasome plays a pivotal role in the pathogenesis of neurodegenerative diseases.

## 2. Alzheimer’s Disease

AD is a highly prevalent form of dementia characterized by the accumulation of extracellular amyloid beta (Aβ) plaques in the brain, neuronal cell death, and neuroinflammation. The immunological scenario of AD-associated neuroinflammation includes an increased production of pro-inflammatory cytokines, a reduced activity of Treg lymphocytes, and the dysregulation of immune-mediated mechanisms of tolerance. Activation of the NLRP3 inflammasome is strongly suggested to play a role in AD-associated neuroinflammation, as shown by results indicating that the concentration of IL-1β and IL-18 is increased in this disease [16,17,18,19].

The NLRP3 inflammasome activation by fibrillar Aβ was initially described in 2008 by Halle et al., who demonstrated that a concentration of 5 mM of fibrillar Aβ induced the production of IL-1β in a NLRP3- and ASC-dependent manner [20]. This result was confirmed by Heneka and colleagues, who showed that NLRP3-deficiency in the transgenic APP/PS1 double-transgenic mouse models of AD, which overexpressed mutated forms of the gene for human amyloid precursor protein (APP) and presenilin 1 (PS1), decreased neuroinflammation, reducing Aβ accumulation and improving neuronal function [21]. Results from other groups showed that NLRP3 or caspase-1 deletion in APP/PS1 mice promoted the differentiation of microglia to an anti-inflammatory M2 phenotype, with decreased secretion of caspase-1 and IL-1β [22]. Further support to the involvement of the NLRP3 inflammasome in the pathogenesis of AD was offered by results showing that Aβ induced the processing of pro-IL-1β into mature IL-1β in the microglia via activation of NLRP3 inflammasome [23].

The NLRP3 inflammasome may, however, not be the only inflammasome that contributes to the pathogenesis of AD [24,25]. Thus, Kaushal and colleagues showed that NLRP1 mRNA was increased in AD neurons and colocalized with caspase-6. Notably, these authors demonstrated that the NLRP1-caspase-1-caspase-6 pathway was involved in the accumulation of Aβ_42_ in serum-deprived neurons [26]. Additional results indicating that NLRP1 mRNA was significantly increased in sporadic and familial AD hippocampal and cortical neurons offer further support to the hypothesis that the NLRP1 inflammasome plays an important pathogenetic role in AD [26].

It is important to underline that, in addition to the inflammatory responses mediated by reactive astrocytes and by activated microglia in the CNS, an activation of the peripheral immune response is also observed and is suggested to contribute to neuroinflammation [27,28,29]. Such peripheral immune response is possibly an attempt to contrast the formation or the extension of Aβ plaques [30,31,32,33,34,35,36,37,38,39] and is associated with the stimulation of peripheral monocytes that are recruited to the CNS. Once these cells reach the CNS, though, they can contribute to the activation of the NLRP3 inflammasome [15]. To further underline the complexity of the immunological impairment that accompanies AD, recent results suggested that alterations of the microbiota might be involved in the generation of the AD-associated inflammatory milieu. Thus, a positive correlation between abundance of the inflammatory bacteria belonging to the taxon Escherichia/Shigellain in stool samples and blood levels of NLRP3 and IL-1β was observed in cognitively impaired elderly individuals with brain amyloidosis [40]. These data thus put forward the hypothesis that dysbiosis results in peripheral inflammation, which is mediated by the activation of the NLRP3 inflammasome and reverberates into the CNS.

Notably, recent in vitro studies conducted in human leukemia monocytic cell line (THP1)-derived macrophage stimulated with Aβ indicated that Leishmania infection down-regulates NLRP3 inflammasome activation, significantly reducing ASC-speck formation, thus favoring the generation of an anti-inflammatory milieu, possibly protecting against AD development [41]. This finding could explain why, despite its strong association with AD risk in industrialized populations, the Apolipoprotein E4 (ApoE4) allele was shown to be associated with improved cognitive functions in members of remote Amazonian tribes. In these populations, the E4 allele has been demonstrated to confer survival in response to infection by parasites that, in turn, could reduce inflammation by reducing the activation of NLRP3 [42].

## 3. Multiple Sclerosis

MS is an autoimmune demyelinating disease of the CNS characterized by immune cell infiltration from the periphery into the CNS as well as by the activation of the microglia and astrocytes, which together promote neuroinflammation and neurodegeneration [43]. A number of studies have suggested the involvement of the NLRP3 inflammasome in the pathogenesis of MS. Gris and colleagues, in 2010 [44], were the first to suggest the critical role of *Nlrp3* gene in the development of experimental autoimmune encephalomyelitis (EAE), the most commonly used experimental model for human MS [45]. Results from their study showed that the absence of *Nlrp3* gene resulted in diminished Th1 and Th17 encephalitogenic responses [44]. In line with this evidence, Peelen et al. reported that the expression level of the inflammasome-related genes NLRP3, IL-1β, and caspase-1, was increased in peripheral blood mononuclear cell (PBMC) from relapsing-remitting (RR) MS patients compared to healthy controls [46].

Results from other groups showed the up-regulation of caspase-1 and IL-1β proteins in PBMCs and cerebrospinal fluid (CSF) of MS patients [47,48]. Moreover, caspase-1 expression was shown to be elevated in MS plaques and PBMC of MS patients [49,50]; taken together these observations lead to the proposal of using serum caspase-1 and ASC protein concentrations as candidate biomarkers for MS onset [51]. IL-18 concentration was observed to be augmented, as well, in serum, CSF, and PBMCs of MS patients [44,52,53,54]. Furthermore, a study by de Jong et al. showed that the increase of IL-1β in CSF was concomitant with a depletion of the IL-1 receptor antagonist (IL-1Ra), an anti-inflammatory protein that antagonizes the binding of IL-1β to its receptor [55]. An indirect support to the role played by IL-1β—a prototypical NLRP3 inflammasome activation-derived cytokine—in the pathogenesis of MS stems from the observation that successful treatment of disease relapses in MS patients with glatiramer acetate or IFNβ results in the increase of endogenous IL-1Ra concentration [56,57]. Notably, IL-18 and IL-1β promote, respectively, IFNγ and IL-17 production by Th1 cells and Th17 cells, two functional T helper lymphocyte subsets that we repeatedly described to play a pivotal role in MS pathogenesis.

The canonical NLRP3 inflammasome requires caspase-1 activation for IL-1β and IL-18 processing. Recent results nevertheless indicated that T cell intrinsic inflammasome activity could drive IL-1β and IL-18 production via caspase-8 activation independently from caspase-1 activation [58,59]. Recent results reinforced a central role for the NLRP3/caspase-8 inflammasome pathway in MS by showing that stimulation of PBMCs from primary progressive MS (PPMS) patients with Monosodium Urate Crystals (MSU) resulted in a significant increase in the expression of NLRP3 and ASC-speck protein and in IL-18 and caspase-8 production. The NLRP3/caspase-8 inflammasome pathway is activated in PPMS, possibly as a consequence of hyperuricemia. Thus, levels of uric acid are upregulated in the CSF of MS patients [60], and the serum uric acid level in patients is potentially associated with susceptibility of MS [61]. Taken together, these results support the hypothesis of hyperuricemia as a common detrimental condition that characterizes MS via the activation of the NLRP3/caspase-8 inflammasome pathway [62].

Finally, the expression of P2X7R, a purinergic receptor that detects and amplifies the release of ATP and, as a consequence, the activation of NLRP3 inflammasome, was shown to be elevated in spinal cords of MS patients [63,64]. In line with this evidence, other studies have shown an association between gain-of-function single nucleotide polymorphisms in the P2X7 receptor gene and MS [65]. On the other hand, glatiramer acetate, one the immunomodulator drugs used for MS, was shown to reduce P2X7R expression [66], suggesting the contribution of extracellular ATP to the pathogenesis of MS. Taken together, these results seem to suggest that endogenous metabolic danger signals, ATP, and uric acid are likely to all be involved in the activation of the NLRP3 inflammasome pathway observed in MS.

## 4. Parkinson’s Disease

PD is a progressive neurodegenerative disorder characterized by the depletion of dopaminergic (DA) neurons in the substantia nigra (SN) and by the accumulation of cytoplasmatic inclusions of fibrillar α-synuclein (α-syn), also called Lewy bodies [67]. Different intracellular mechanisms allow the release of α-syn outside of the cell [68], but the common endpoint of α-syn accumulation is the activation of astrocytes and microglia to produce IL-1β [68,69]. Notably, this phenomenon also facilitates the recruitment of immune cells from the periphery into the CNS [70].

A possible involvement of the NLRP3 inflammasome in the pathogenesis of PD was initially suggested in 2013 by Codolo et al., who demonstrated in vitro that, while both monomeric and fibrillary α-syn increased pro-IL-1β levels via toll-like receptor (TLR)-2 signaling, the fibrillary form of α-syn, alone, stimulated the inflammasome by activating caspase-1, resulting in IL-1β production [71]. The result of this study was further confirmed in vivo using an animal model of PD. Thus, injections of neurotoxin 1-methyl-4-phenyl-1,2,3,6-tetrahydropyridine (MPTP) caused the loss of dopaminergic neurons in the substantia nigra and a PD-like pathology. *Nlrp3* deficient mice were shown to be resistant to PD, strongly suggesting an important role of the NLRP3 inflammasome in the pathogenesis of PD [72]. Further results showed that miR-7, a microRNA known to regulate α-syn gene expression [73], was present in the midbrain of the MPTP-induced PD mice model. Notably, as *Nlrp3* is a target gene of miR-7, the stereotactic injection of miR-7 mimics in the mouse brain was demonstrated to inhibit the NLRP3 inflammasome activation, reducing neuroinflammation [74]. Additional results indicated that the exogenous administration of IL-1Ra attenuated the MPTP-induced PD phenotypes in mice [75]. Further support to the possible role of the NLRP3 inflammasome in PD is based on the knowledge that dopamine neurons negatively regulate NLRP3 by the dopamine D1 receptor (DRD1)-cyclic adenosine monophosphate (cAMP) signaling pathway. Based on this information, it was observed that DRD1−/− mice were less resistant to MPTP-induced neuroinflammation, as shown by the increased IL-18 and IL-1β production and the more extensive damages inflicted on dopaminergic neurons [76].

Whereas the overall consensus of results obtained in the animal model strongly supports an association between the NLRP3 inflammasome and PD, data stemming from analyses performed in patients with a diagnosis of PD are much less convincing. To summarize, in PD patients compared to healthy controls, (1) CSF concentration of IL-1β and IL-18 was found to be higher [77]; (2) serum concentration of IL-1β as well as caspase-1 activity were shown to be increased [78]; and (3) protein levels of NLRP3, caspase-1, and IL-1β were seen to be augmented in PBMCs [78]. Nevertheless, results from other groups showed that, whereas NLRP3 serum levels were increased in PD patients compared to healthy controls (HC), no differences in IL-1β and IL-18 serum levels could be detected [77]. Even more recently, we observed that stimulation of PBMC with monomeric or aggregated α-syn induced a comparable NLRP3 and ASC-speck expression, as well as IL-18 and caspase-1 production in cells of PD patients and healthy controls, indicating that α-syn does not stimulate the NLRP3 inflammasome activity. Interestingly, IL1β and IL-6 production was increased, whereas that of IL-10 was reduced in α-syn-stimulated cells of PD patients, suggesting that PD-associated neuroinflammation is not the consequence of the activation of the NLRP3 inflammasome but rather of an imbalance between pro- and anti-inflammatory cytokines.

In conclusion, although several studies have shown that α-synuclein can elicit activation of inflammasome in monocyte and microglial cell lines and in PD animal models, the possible role of NLRP3 in patients with a diagnosis of PD still needs to be clarified.

## 5. Amyotrophic Lateral Sclerosis

ALS is a neurodegenerative disease characterized by the selective loss of motor neurons in the motor cortex, the brainstem, and the spinal cord. The vast majority of ALS cases are sporadic, (sALS), but a small fraction (about 5–10%) of cases are familiar (fALS). In this situation, mutations in a number of genes, the most frequent of which is the mutation in Cu^2+^/Zn^2+^ superoxide dismutase (SOD1), are known to associate with the disease.

Increasing evidence has proposed an important role for neuroinflammation in the pathogenesis of ALS, as demonstrated by the infiltration of lymphocytes and macrophages in the CNS, the activation of the microglia, and the presence of reactive astrocytes in the same anatomical sites where motor neuron injuries are observed. Recent studies have suggested that a dysregulated and excessive inflammasome activation contributes to the neuroinflammation observed in ALS [79,80]. Thus, data obtained in the G93A-*SOD1* transgenic mice, the most common animal model for ALS, showed the activation of caspase-1 and IL-1β in the microglia by ALS-linked mutant SOD1 and demonstrated that caspase-1 or IL-1β genes knockout or the use of recombinant IL-1Ra resulted in a reduction of inflammation. Notably, augmented caspase-1 and IL-1β production appeared to be NLRP3-independent in this model, suggesting the possible involvement of other inflammasome complexes [81]. In partial contrast with these results, other analyses performed in the *SOD1* transgenic mice showed an upregulation of NLRP3 and ASC in the anterior dorsal thalamic nucleus (AD) of G93A- [82] and of the transactive response DNA-binding protein-43 (TDP-43) in the microglia [83]. Other results showed that in G93A-*SOD1* transgenic mice and in human tissues, spinal cord astrocytes were activated, expressed NLRP3-inflammasome proteins, and contributed to inflammation in ALS by releasing proinflammatory cytokines [84]. In the same work, the authors noticed the microglial expression of ASC but not that of NLRP3, suggesting that other inflammasome sensor molecules may play a role in microglia-driven neuroinflammation in ALS.

Fewer results are available in humans; in ALS patients, serum concentration of IL-18, but not of IL-1β, was observed to be increased [85], and NLRP3 and caspase-1 expression was shown to be augmented in brain tissues [86]. It is also important to underline that clinical studies using the Interleukin-1 receptor antagonist Anakinra have not demonstrated a reduction in the neuroinflammation in ALS, suggesting that NLRP3 inflammasome might not play a major role in ALS or that this disease is mainly driven by IL-18 and not IL-1β [87]. To further augment the uncertainty of the possible role of the inflammasome in ALS, data show that 17β-estradiol, a steroid hormone that down-regulates inflammasome activation, improves motor neuron survival in a humanized animal model of ALS that carries the human SOD1 (G93A) mutation [88]. However, a conclusion has still not been reached; extensive analyses will be needed to dissect the possibility that the inflammasome is involved in the pathogenesis of ASL.

Literature data concerning expression of inflammasome proteins and effector cells in neurodegenerative disease are summarized in Table 1.

## 6. Pharmacological Modulation of the Inflammasome

Given the role of the NLRP3 inflammasome in neuroinflammation, a number of studies has been conducted in the exploration of possible therapeutic pathways for neurodegenerative diseases through the inhibition of the NLRP3 inflammasome.

To date, NLRP3 inhibitors can be divided into those that directly inhibit NLRP3 or those that mediate NLRP3 inactivation as a consequence of the inhibition of inflammasome components or related signaling events. Available compounds act mainly by inhibiting the products of inflammasome activation, for example, by impeding the biological effects of IL-1β via the use of either anti-IL-1β antibodies or IL-1Ra; notably, no effective anti-IL-18 therapies are currently available [89]. Three biologics are approved by the US Food and Drug Administration (FDA) for multiple inflammatory diseases: canakinumab, an IL-1β- neutralizing antibody; anakinra, a recombinant IL-1 receptor antagonist; and rilonacept, a decoy receptor that binds IL-1β and IL-1α. Nevertheless, although their efficacy has been demonstrated for autoinflammatory diseases, there no report of their use in clinical trials in neurodegenerative diseases [90]. Because IL-1β production can be mediated by other inflammasomes, specific inhibitors that directly target the NLRP3 inflammasome could be a better option for treatment of diseases in which inflammation is the consequence of NLRP3 activation.

Some compounds have shown an inhibitory effect in vitro on NLRP3 inflammasome activation, including MCC950 [91], β-hydroxybutyrate (BHB) [92], Bay 11- 7082 [93], dimethyl sulfoxide (DMSO) [94], and type I interferon [95]. However, most of these inhibitors are relatively nonspecific and have low efficacy. Among the direct NLRP3 inhibitors, the diarylsulfonylurea compound MCC950 (originally reported as CRID3/CP-456773) is the most potent and specific for NLRP3. MCC950 demonstrated therapeutic efficacy against several preclinical immunopathological models, including EAE [91], AD [22], and PD [96]. However, this compound is currently not approved by the FDA for the therapy of neurodegenerative disease.

Using a different approach, Stavudine (d4T), an antiviral nucleoside reverse transcriptase inhibitor (NRTIs) designed to target HIV, was recently shown to down-modulate NLRP3 inflammasome activation in mice [97]. Additional data confirmed the ability of this compound to hamper NLRP3 inflammasome activation in an in vitro model of AD by reducing NLRP3 assembly as well as IL-18 and caspase-1 production and stimulating amyloid-beta autophagy by macrophages [18].

Given the lack of effective drugs in the therapy of chronic neurodegenerative conditions and the role of NLRP3 inflammasome in the pathogenesis and progression of these diseases, efforts should be made to develop effective therapeutic strategies, possibly including those targeting the NLRP3 inflammasome.

## Figures and Tables

**Figure 1 molecules-26-00953-f001:**
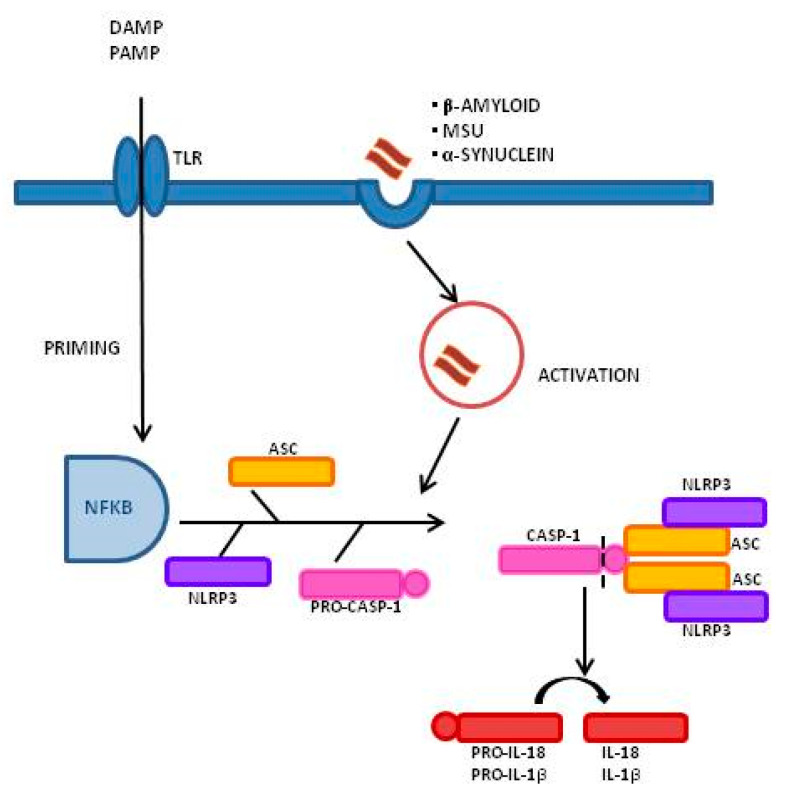
Inflammasome activation process and signaling mechanism: A Two-Signal Model for NLRP3 Inflammasome Activation. The priming signal (signal 1, left) is provided by damage-associated molecular patterns (DAMP) or by pathogen-associated molecular patterns (PAMP), leading to the activation of the transcription factor NF-κB and subsequent upregulation of NLRP3, pro-interleukin-1β (pro-IL-1β), and pro-interleukin-IL-18 (pro-IL-18). The activation signal (signal 2, right) is provided by a variety of stimuli including β-amyloid, α-synuclein, and Monosodium Urate Crystals (MSU), leading to the assembly and formation of the NLRP3 inflammasome through the combination of NLRP3, ASC, and procaspase-1, and leading to the production of caspase-1, which catalyzes the transformation from pro-IL-1β and pro-IL-18 into IL-1β and IL-18.

**Table 1 molecules-26-00953-t001:** Changes in levels of inflammasome components and effectors in cells and tissues from human samples in neurodegenerative diseases.

Disease	System	Inflammasome Component/Effector	References
Alzheimer’s	Human	↑ IL-1β, ↑ IL-18 ↑ NLRP1, ↑ ASC, ↑ caspase-1 ↑ NLRP3, ↑ IL-1β	Awad et al., 2017 [16] Heneka et al., 2013 [17] La Rosa et al., 2017 [18] Saresella et al., 2017 [19] Kaushal et al., 2015 [26] Cattaneo et al., 2016 [40]
Multiple Sclerosis	Human	↑ NLRP3, ↑ IL-1β, ↑caspase-1 ↑ caspase-1, ↑ IL-1β ↑ caspase-1 ↑ IL-18 ↑ IL-1β ↑ NLRP3, ↑ ASC, ↑ IL-18, ↑ caspase-8	Peleen et al., 2015 [46] Inoue & Shinohara, 2013 [47] Mamik&Power, 2017 [48] Furlan R et al., 1999 [49] Ming X et al., 2002 [50] Losy J et al., 2001 [52] Nicoletti F et al., 2001 [53] Chen YC et al., 2012 [54] Gris D et al., 2010 [44] de Jong et al., 2002 [55] Piancone et al., 2018 [62]
Parkinson’s Disease	Human	↑ IL-1β ↑ IL-1β, ↑ IL-18 ↑ IL-1β, ↑ caspase-1 ↑ NLRP3, ↑ IL-1β, ↑ caspase-1	Codolo et al., 2013 [71] Zhang et al., 2016 [77] Zhou Y et al., 2015 [74] Fan et al., 2020 [78]
Amyotrophic Lateral Sclerosis	Human	↑ IL-18 ↑ NLRP3, ↑ ASC, ↑ IL-18, ↑ caspase-1 ↑ NLRP3, ↑ caspase-1	Italiani et al., 2014 [85] Johann et al., 2015 [84] Kadhim et al., 2016 [86]

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
