# Peer review of "The Role of the Inflammasome in Neurodegenerative Diseases"

_molecules, 2021, doi:10.3390/molecules26040953_

Round 1
Reviewer 1 Report
The authors summarize results identifying multiprotein complexes, so called inflammasomes, involved in the activation pathway of inflammatory responses, developed in the course of neurodegenerative disease. The review is interesting, even though there is a bias in interpreting “the inflammasome as a pivotal player in the pathogenesis of neurodegenerative diseases”.
While inflammasome can play a pivotal role in the secondary neurodegeneration associated with multiple sclerosis, it is generally accepted that primary neurodegenerative diseases like Alzheimer Disease are caused by specific proteinopathies, wherein inflammasome and inflammation can just represent a secondary event to the accumulation of misfolded proteins. The authors should add some criticism to their enthusiasm for the theory on the role of inflammasome in the pathogenesis of neurodegenerative diseases, specifying that actually there are not proves that the inflammatory responses are excessive, neither that inflammation is the primary cause of protein misfolding in neurodegenerative diseases.
Author Response
Thank you. On the basis of your suggestion we discuss this aspect and add in the text the following consideration:
Nevertheless, it has to be considered that in primary neurodegenerative diseases, characterized by the accumulation of misfolded proteins, like AD and PD, it is not clear if inflammation might be the primary cause of disease or, possibly, a reaction to pathology. Indeed, the pathophysiological hypothesis of neurodegenerative diseases relies on the fact that some proteins, changing their conformations, aggregate into fibrils or oligomers resulting in neurotoxicity and leading to neurodegeneration and inflammation [4-7].
- Ciccocioppo, F.; Bologna, G.; Ercolino, E.; Pierdomenico, L.; Simeone, P.; Lanuti, P.; Pieragostino, D.; Del Boccio, P.; Marchisio, M.; Miscia, S. Neurodegenerative diseases as proteinopathies-driven immune disorders. Neural. Regen. Res. 2020, 15(5), 850-856. doi:10.4103/1673-5374.268971.
- Bayer, T.A. Proteinopathies, a core concept for understanding and ultimately treating degenerative disorders? Eur. Neuropsychopharmacol. 2015, 25(5), 713-24. doi: 10.1016/j.euroneuro.2013.03.007.
- Sami, N.; Rahman, S.; Kumar, V.; Zaidi, S.; Islam, A.; Ali, S.; Ahmad, F.; Hassan, M.I. Protein aggregation, misfolding and consequential human neurodegenerative diseases. Int. J. Neurosci. 2017, 127:1047-1057. doi: 10.1080/00207454.2017.1286339.
- Soto, C.; Pritzkow, S. Protein misfolding, aggregation, and conformational strains in neurodegenerative diseases. Nat. Neurosci2018; 21: 1332-1340. doi: 10.1038/s41593-018-0235-9.
Reviewer 2 Report
The manuscript of Frederica Piancone et al., is clearly written and addresses basic aspects of all mentioned neurodegenerative diseases in relation to inflammatory activation. The review is descriptive and clearly characterizes involvement of NLRP3 in inflammatory response in individual disease but does not address any identified or at least hypothetical abnormalities in NLRP3 regulation, expression, or activation which could be disease specific, leading to clinical manifestation in only affected subjects group.
Please check all abbreviations to be explained in the text (ALR, APP/PS1).
In the Table 1 Changes in levels of inflammasome components and effectors in cells and tissues from human samples in neurodegenerative diseases the column "human" according to title.
The statement in the sentence on lines (128-131) should be more precisely formulated explained.
Author Response
Response 2
- Thank you for your suggestion. All the abbreviations have been checked and explained in the text, that has been modified accordingly.
- The column “human” has been modified according to the title.
- As it regards the statement in the sentence on lines 128-131
“This finding could explain why despite its strong association with AD risk in industrialized populations, the Apolipoprotein E4 (ApoE4) allele was shown to be associated with improved cognitive functions in members of remote Amazonian tribes, provided that they are infected with parasites [38]”
Thank you for your suggestion. The sentence has been reformulated as follows:
“This finding could explain why despite its strong association with AD risk in industrialized populations, the Apolipoprotein E4 (ApoE4) allele was shown to be associated with improved cognitive functions in members of remote Amazonian tribes. In these populations the E4 allele, often associated with cognitive decline in industrialized populations, has been demonstrated to confer survival in response to infection by parasites, that, in turn, could reduce inflammation by reducing the activation of NLRP3 [38]”.
Round 2
Reviewer 2 Report
I am satisfied with authors response.